# IDEA: An Invariant Perspective for Efficient Domain Adaptive Image Retrieval

**Haixin Wang[1,\*], Hao Wu[2,\*], Jinan Sun[1], Shikun Zhang[1],**
**Chong Chen[1], Xian-Sheng Hua[3], Xiao Luo[4,†]**
[1]Peking University, [2]University of Science and Technology of China,
[3]Terminus Group, [4]University of California, Los Angeles
wang.hx@stu.pku.edu.cn, wuhao2022@mail.ustc.edu.cn,
sjn@pku.edu.cn, zhangsk@pku.edu.cn, chenchong.cz@gmail.com,
huaxiansheng@gmail.com, xiaoluo@cs.ucla.edu

## Abstract

In this paper, we study the problem of unsupervised domain adaptive retrieval, which transfers retrieval models from a label-rich source domain to a label-scarce target domain. Although there exist numerous approaches that incorporate transfer learning techniques into deep hashing frameworks, they often overlook the crucial invariance needed for adequate alignment between these two domains. Even worse, these methods fail to distinguish between causal and non-causal effects embedded in images, making cross-domain retrieval ineffective. To address these challenges, we propose an Invariance-acquired Domain Adaptive Hashing (IDEA) model. Our IDEA first decomposes each image into a causal feature representing label information and a non-causal feature indicating domain information. We then generate discriminative hash codes using causal features with consistency learning on both source and target domains. More importantly, we employ a generative model for synthetic samples to simulate the intervention of various non-causal effects, thereby minimizing their impact on hash codes for domain invariance. Comprehensive experiments conducted on benchmark datasets confirm the superior performance of our proposed IDEA compared to a variety of competitive baselines.

## 1 Introduction

Approximate nearest neighbor (ANN) search [73, 77, 49, 29, 40] has been a pivotal research problem within the field of deep learning for decades. Recently, learning to hash [41, 80, 68, 20] has come to the fore in ANN search engines due to its high computational and memory efficiency. The main objective of learning to hash is to convert high-dimensional data points into compact low-dimensional binary codes while maintaining semantic similarity. Consequently, this approach mitigates the requirement for computationally expensive pairwise distance computations, replacing them with economical Hamming distance calculations using bit-wise XOR and bit-count operations [60].

Recent hashing techniques can primarily be classified into two types, namely supervised approaches [72, 44, 67, 48, 75, 8, 31, 52] and unsupervised approaches [32, 76, 46, 69, 33, 61, 10]. Supervised hashing approaches incorporate annotation data into the similarity learning algorithms, generally leading to better performance than unsupervised ones. These supervised methods primarily focus on minimizing a spectrum of similarity-preserving losses and quantization losses [11, 78] to generate highly discriminative hash codes. Furthermore, several innovative proxy-based techniques have been introduced where a proxy is constructed in the Hamming space for each class, and hash

---

*Equal contribution. †Corresponding author.

codes are enforced to approximate their respective proxies. These methods have demonstrated encouraging results in efficient image retrieval.

However, these supervised approaches rely on an assumption that the distributions between the training and query data are the same, which is often violated in real-world scenarios. For instance, when an ANN search engine trained on benchmark images, is employed to search query images from mobile phones, retrieval performance could potentially diminish substantially [45, 22, 64]. In light of this, this paper concentrates on the problem of domain adaptive retrieval, which aims to enhance the performance of retrieval systems by leveraging both labeled source instances and unlabeled target samples. In literature, a variety of domain adaptive hashing methods have been proposed [71, 66, 64, 47, 74, 21, 58], which typically incorporate domain adaptation strategies into deep hashing frameworks. They adopt the memory bank [64] and adversarial learning [58, 36] to achieve alignment across source and target domains. Moreover, pseudo-labeling [19, 58] is implemented to extract knowledge from unlabeled target samples, thus addressing the challenge of label scarcity in the target domain.

Unfortunately, existing approaches have two intrinsic limitations that hinder their effectiveness as follows: (1) *Entanglement of causal and non-causal effects.* Current methods tend to recklessly focus on correlations between image data and semantic labels. This could result in an entanglement of causal and non-causal factors. Consequently, the hash codes derived from these deep features may lack interpretability and be suboptimal for subsequent retrieval tasks. (2) *Neglect of invariant patterns.* While adversarial learning is commonly used to implicitly align hash codes between source and target domains, these methods often fail to explicitly capture the invariant patterns present in vision representations. This could lead to ineffective domain alignment of hash codes due to the influence of underlying variant factors. As a result, an effective hashing framework under the principle of invariant learning [2, 5] is highly desired.

In this paper, we explore domain adaptive hashing through a causal analysis lens, proposing a model named Invariance-Acquired Domain Adaptive Hashing (IDEA) for efficient image retrieval under domain shift. The core of domain adaptive hashing lies in discerning causal and non-causal effects within images. To generate high-quality and interpretable binary codes, we disentangle causal and non-causal features within each image, guided by the principles of the information bottleneck. On one hand, we maximize the mutual information between causal features and label information. On the other hand, we retain the most information in hidden features for non-causal features while minimizing the information in labels. These causal features are used to generate discriminative hash codes through consistency learning across both source and target domains. To further mitigate the non-causal effects, we incorporate a generative model that simulates the intervention of various non-causal effects, thereby encouraging hash codes to remain sufficiently invariant to different non-causal components. Leveraging invariant learning on causal effects, our IDEA is capable of generating domain-invariant hash codes that facilitate efficient cross-domain retrieval. Empirical experiments conducted on a variety of benchmark datasets validate the superior performance of our IDEA compared to competitive baselines. The contribution of this paper is summarized as follows:

- *Problem Connection.* We pioneer a novel perspective that connects invariant learning with domain adaptive hashing for efficient image retrieval.

- *Novel Methodology.* Our IDEA not only disentangles causal and non-causal features in each image following the principle of the information bottleneck, but also ensures hash codes are sufficiently invariant to the intervention of non-causal features.

- *High Performance.* Comprehensive experiments across numerous datasets demonstrate that our IDEA outperforms a range of competitive baselines in different settings.

## 2 Related Works

**Learning to Hash.** The task of efficient image retrieval has recently witnessed significant interest [49, 29, 40, 37], leading to the development of deep hashing methods [41, 80, 68, 20], including both supervised [72, 44, 67, 48, 75, 8, 31, 52, 54] and unsupervised approaches [32, 76, 46, 69, 33, 61, 56, 10, 55]. Deep unsupervised hashing methods typically reconstruct semantic structures using similarity metrics, subsequently generating similarity-preserving binary codes. Self-supervised methods have also been developed to enhance the performance. The incorporation of label information has notably improved the performance of deep supervised hashing methods. Early attempts at optimizing the

hashing network often involved pairwise and triplet losses based on similarity structures [44, 67, 30, 4]. Another line of study has utilized point-wise optimization, establishing proxies for each category within the Hamming space and compelling hash codes to approximate these proxies [72, 13]. However, these methods typically overlook potential distribution shift in practical scenarios, which can significantly degrade retrieval performance. This limitation has prompted research into domain adaptive hashing.

**Unsupervised Domain Adaptation.** Unsupervised Domain Adaptation (UDA) has long been a formidable challenge in machine learning and computer vision [38, 51, 14]. Early methods often explicitly reduce distribution discrepancies for domain alignment [43, 27]. An alternative approach utilizes adversarial learning for implicit domain alignment, incorporating a gradient reversal layer and a domain discriminator to engage in a minimax game [15, 36]. Recently, researchers have extended this topic towards efficient domain adaptive retrieval [71, 66, 64, 47, 74, 21, 58, 59], proposing a number of transferable hashing methods to address this problem. These methods typically integrate domain adaptation models into deep hashing frameworks and generate similarity structures for learning semantic knowledge on the target domain. Despite these advancements, these methods often fall short due to the entanglement of causal and non-causal effects and the neglect of invariant patterns. In response to these challenges, we propose an effective approach IDEA.

**Invariant Learning.** Invariant learning [2, 5, 9] aims to identify invariant correlations between inputs and targets under domain shift, while concurrently eliminating spurious and variant relationships. This concept has been explored in the context of out-of-distribution generalization. Under specific assumptions, invariant learning has demonstrated significant potential for model generalization following causal theory. Invariant risk minimization [2] (IRM) has been proposed to regularize neural networks to remain stable under the environmental variance, showing superior performance when compared to empirical risk minimization [12] (ERM). Moreover, MVDC [65] learns from frequency spectrum to generate causal features for domain generalization for cross-domain object detection. Invariant learning has also been extended to address challenges in graph domains [28, 70, 6, 53, 63]. For instance, CIGA [6] leverages distinct subgraphs for graph contrastive learning, thereby achieving superior out-of-distribution graph classification performance. Our study establishes a connection between invariant learning and domain adaptive hashing, generating domain-invariant and discriminative hash codes for cross-domain image retrieval. To the best of our knowledge, this is the first work to apply invariant learning for transferable retrieval tasks, and is proven effective for potential application.

# 3 Problem Definition

Given a source domain $\mathcal{D}^s = \{(\boldsymbol{x}_i^s, y_i^s)\}_{i=1}^{N_s}$ with $N_s$ fully-labeled images and a target domain $\mathcal{D}^t = \{(\boldsymbol{x}_j^t)\}_{j=1}^{N_t}$ with unlabeled $N_t$ images, both domain share a common label space $\mathcal{Y} = \{1, 2, \cdots, C\}$ despite potential distribution shifts. The objective is to develop a hashing-based retrieval model that projects an input image $\boldsymbol{x}$ onto a compact binary code $\boldsymbol{b} \in \{-1, 1\}^L$, where $L$ represents the code length. This model should ensure that similar images are mapped to comparable binary codes within the Hamming space, and its performance should be evaluated in both single-domain and cross-domain retrieval systems. In a single-domain retrieval system, both query and database images originate from the target domain, whereas, in a cross-domain retrieval system, the query images are drawn from the target domain $\mathcal{D}^t$ and the database images are sourced from the source domain $\mathcal{D}^s$.

# 4 Method

## 4.1 Framework Overview

We address the problem of unsupervised domain adaptive retrieval. While several domain adaptive hashing algorithms have been proposed, they often neglect invariant learning during the hash code generation process. This oversight often leads to significant domain discrepancy in the resulting hash codes. More- over, these methods indiscriminately focus on correlations between image data and semantic labels,

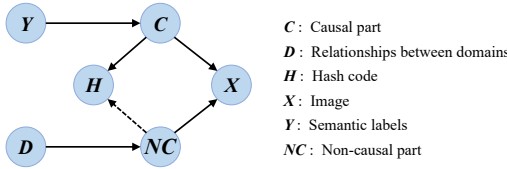

$C$ : Causal part
$D$ : Relationships between domains
$H$ : Hash code
$X$ : Image
$Y$ : Semantic labels
$NC$ : Non-causal part

Figure 1: Structural causal model in our problem.

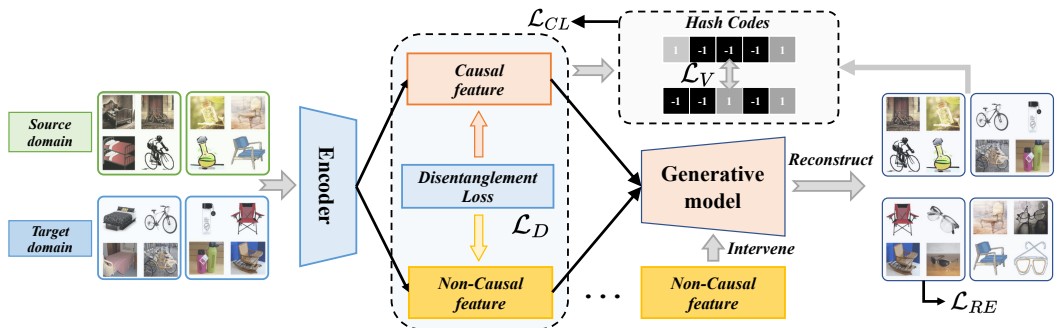

Figure 2: The framework of the proposed IDEA. We feed both source and target images into an encoder. Then each image is disentangled into causal features and a non-causal features. The causal features are adopted to generate hash codes. A generative model is utilized to reconstruct the original graphs. We add intervention by using a different non-causal features and minimize the domain shift by invariant learning.

failing to disentangle the causal and non-causal effects encoded in images. This results in learnt hash codes that are not only suboptimal for downstream retrieval tasks, but are also difficult to interpret.

In this study, we present a novel approach named IDEA, to tackle these issues. The fundamental idea is to incorporate invariant learning into the optimization of the hashing network. Specifically, we first introduce a structural causal model, as illustrated in Figure 1, and analyze the correlation within this problem. We then separate each image into causal and non-causal features using an information bottleneck. The causal features are subsequently used for hash code production. Lastly, we introduce interventions on non-causal effects and limit their influence on invariant hash codes. More details can be found in Figure 2.

## 4.2 Structure Causal Model

To start, we apply a causal perspective to domain adaptive hashing and construct a structural causal model [39] (SCM) to depict the image generation process under domain shifts. This model is used to illustrate the relationships between domains $D$, semantic labels $Y$, causal parts $C$, non-causal parts $NC$, images $X$, and hash codes $H$. The specific relationships are elaborated as follows:

- $Y \to C$. Semantic labels produce the causal part which should be invariant to domain distributions.

- $D \to NC$. Domain information provides the non-causal part which is changeable across domains.

- $C \to X \leftarrow NC$. Each image is generated by combining both causal part and non-causal part.

- $NC \dashrightarrow C \to H$. To obtain domain-invariant hash codes, we should generate hash codes using causal features. However, there could be additional relationships between $C$ and $NC$. For example, $C$ and $NC$ can be simultaneously influenced by a cause. Therefore, $C$ acts as a confounder between $NC$ and $H$, i.e., resulting in falsely related variables.

The essence of domain adaptive hashing lies in learning domain-invariant and discriminative binary codes. From the structural causal model, we need to construct mappings $\phi(\cdot)$ and $\psi(\cdot)$ which fulfill the following conditions:

$$[C, NC] = \phi(X), H = \psi(C), H \perp NC \mid C, \tag{1}$$

where $\perp$ symbolizes the independence under the given condition. Inferred from Eqn. 1, we are required to address the following challenges: 1) The first term necessitates us to disentangle the causal and non-causal components of each image; 2) The second term requires us to learn discriminative hash codes from causal components; 3) final term advocates for the generation of hash codes that are independent of the influence of non-causal components. With these goals in mind, we propose a novel deep hashing method IDEA for domain adaptive retrieval.

## 4.3 Causality-acquired Disentanglement

In this part, we apply the information bottleneck principle [1] to disentangle each image into causal and non-causal features. This approach facilitates the extraction of multifaceted latent factors embedded within images, enabling the generation of high-quality and interpretable binary codes. As suggested by the structural causal model, causal features should encapsulate label information, while non-causal features should reflect domain discrepancy.

In detail, we first introduce a feature extractor $F(\cdot)$, which removes the last classification layer in a popular neural network backbone (e.g., VGG-F [50]) to generate hidden features, i.e., $\boldsymbol{z} = F(\boldsymbol{x})$. Then, two different MLPs $g^c(\cdot)$ and $g^n(\cdot)$ are utilized to generate two features, respectively, i.e., $\boldsymbol{f}^c = g^c(\boldsymbol{z})$ and $\boldsymbol{f}^n = g^n(\boldsymbol{z})$. Let $\boldsymbol{F}^c$ and $\boldsymbol{Y}$ denote the random variables of causal features and semantic labels, and we maximize the mutual information between $\boldsymbol{F}^c$ and $\boldsymbol{Y}$. To achieve this, we turn to InfoNCE [18, 3, 7], which construct positive pairs and negative pairs from the joint distribution $p(\boldsymbol{F}^c, \boldsymbol{Y}) \in \mathcal{D}^s$ and the product of marginal distributions $p(\boldsymbol{F}^c)p(\boldsymbol{Y}) \in \mathcal{D}^s$, respectively. Since label information is not available in $\mathcal{D}^t$, we merely utilize source data in this module. In particular, an estimator $T^c$ is introduced to estimate the lower bound of mutual information, the target objective is written as:

$$\min \mathcal{L}_{MI}^c = \mathbb{E}_{p(\boldsymbol{F}^c, \boldsymbol{Y})} \left[ T^c(\boldsymbol{f}^c, \boldsymbol{y}) \right] - \log \left( \mathbb{E}_{p(\boldsymbol{F}^c)p(\boldsymbol{Y})} \left[ e^{T^c(\boldsymbol{f}^c, \boldsymbol{y})} \right] \right), \tag{2}$$

where $T^c(\cdot, \cdot)$ comes from a bi-linear function with a weight matrix $\boldsymbol{W}^c$ to calculate the probability of being a positive pair, i.e., $T^c(\boldsymbol{f}^c, \boldsymbol{y}) = \boldsymbol{f}^c \boldsymbol{W}^c \boldsymbol{y}$. The estimator and the causal head $g^c$ are optimized jointly to obtain effective causal features with high correlation to label information.

Moreover, to generate non-causal features, we minimize the mutual information between $\boldsymbol{F}^n$ and $\boldsymbol{Y}$ with the maximum mutual information with hidden features $\boldsymbol{Z}$ following the principle of information bottleneck. In formulation, we need to minimize:

$$\min \mathcal{L}_{MI}^n = I(\boldsymbol{F}^n, \boldsymbol{Y}) - \beta I(\boldsymbol{F}^n, \boldsymbol{Z}), \tag{3}$$

where $\beta$ is a parameter to balance the loss. Similarly, it is infeasible to minimize Eqn. 3 directly. Therefore, we calculate the upper bound and lower bound of $I(\boldsymbol{F}^n, \boldsymbol{Y})$ and $I(\boldsymbol{F}^n, \boldsymbol{Z})$, respectively. Here, we first define:

$$\hat{I}(\boldsymbol{F}^n, \boldsymbol{Y}) = \mathbb{E}_{p(\boldsymbol{F}^n, \boldsymbol{Y})} \left[ \log p(\boldsymbol{y} \mid \boldsymbol{f}^n) \right] - \mathbb{E}_{p(\boldsymbol{F}^n)} \mathbb{E}_{p(\boldsymbol{Y})} \left[ \log p(\boldsymbol{y} \mid \boldsymbol{f}^n) \right], \tag{4}$$

which can be shown as an upper bound of $I(\boldsymbol{F}^n, \boldsymbol{Y})$. More detail is shown in Appendix. Moreover, we introduce a variational function $q(y|\boldsymbol{f}^n)$ to approximate $p(y|\boldsymbol{f}^n)$. To measure the lower bound of $I(\boldsymbol{F}^n, \boldsymbol{Z})$, we introduce a different estimator $T^n(\boldsymbol{f}^n, \boldsymbol{z}) = \boldsymbol{f}^n \boldsymbol{W}^n \boldsymbol{z}$ and the upper bound of $I(\boldsymbol{F}^n, \boldsymbol{Z})$ is written as follows:

$$\hat{I}(\boldsymbol{F}^n, \boldsymbol{Z}) = \mathbb{E}_{p(\boldsymbol{F}^n, Z)} \left[ T^n(\boldsymbol{f}^n, \boldsymbol{z}) \right] - \log \left( \mathbb{E}_{p(\boldsymbol{F}^n)p(\boldsymbol{Z})} \left[ e^{T^n(\boldsymbol{f}^c, \boldsymbol{z})} \right] \right), \tag{5}$$

By combining Eqn. 16 and Eqn. 5, we rewrite the objective in Eqn. 3 as:

$$\begin{aligned} \mathcal{L}_{MI}^n = \mathbb{E}_{p(\boldsymbol{F}^n, \boldsymbol{Y})} \left[ \log q_\mu(\boldsymbol{y} \mid \boldsymbol{f}^n) \right] - \mathbb{E}_{p(\boldsymbol{F}^n)} \mathbb{E}_{p(\boldsymbol{Y})} \left[ \log q(\boldsymbol{y} \mid \boldsymbol{f}^n) \right] \\ - \beta \left( \sup_{T^n} \mathbb{E}_{p(\boldsymbol{F}^n, \boldsymbol{Z})} \left[ T^n(\boldsymbol{f}^n, \boldsymbol{z}) \right] - \log \left( \mathbb{E}_{p(\boldsymbol{F}^n)p(\boldsymbol{Z})} \left[ e^{T^n(\boldsymbol{f}^n, \boldsymbol{z})} \right] \right) \right), \end{aligned} \tag{6}$$

From Eqn. 6, we utilize adversarial learning to optimize non-causal features, which generates reserve gradients for estimator $T^n$ when minimizing $\mathcal{L}_{MI}^n$. The overall loss for disentanglement is summarized as:

$$\mathcal{L}_D = \mathcal{L}_{MI}^c + \mathcal{L}_{MI}^n. \tag{7}$$

By minimizing $\mathcal{L}_D$, we can successfully disentangle causal and non-causal elements embedded in images, which can guide us to generate high-quality and interpretable hash codes.

## 4.4 Consistency Learning for Discriminative Hash Codes

To learn discriminative hash codes, we leverage causal features given their strong correlation with label information for hash code generation. Here, consistency learning is introduced, based on the principle that similar images should be mapped to similar hash codes to enable effective image retrieval [4, 25]. This strategy ensures that our model can capture and preserve the inherent semantic relationships in the data, thus facilitating more accurate and effective retrieval.

In particular, we utilize an MLP $\psi(\cdot)$ to map causal features to hash codes and enforce similar images to have similar binary codes. Here, we resort to consistency learning for optimization, which constructs positives via (1) two images sharing the same label in the source domain; (2) two augmented views from the same image in the target domain. In practice, we generate two augmented views for each image in a mini-batch $\mathcal{B} = \mathcal{B}^s \cup \mathcal{B}^t$, and define the positive set for source sample as:

$$\Pi(\hat{\boldsymbol{x}}_i^s) = \{k|\hat{y}_i^s = \hat{y}_j^s\}, \tag{8}$$

where $\hat{\boldsymbol{x}}_i^s$ denotes the augmented source sample. Then, we provide a hashing consistency learning objective for source domain:

$$\mathcal{L}_{CL}^s = -\mathbb{E}_{\boldsymbol{x} \in \mathcal{B}^s} \frac{1}{|\Pi(\hat{\boldsymbol{x}}_i^s)|} \sum_{k \in \Pi(\hat{\boldsymbol{x}}_i^s)} \log \frac{\exp\left(\hat{\boldsymbol{b}}_i \cdot \hat{\boldsymbol{b}}_k / \tau\right)}{\sum_{i=1}^{2|\mathcal{B}^s|} \exp\left(\hat{\boldsymbol{b}}_i^s \cdot \hat{\boldsymbol{b}}_{k'}^s / \tau\right)}, \tag{9}$$

where $\hat{\boldsymbol{b}}_i^s = \psi(g^c(F(\hat{\boldsymbol{x}}_i^s)))$ and $\tau$ is a temperature parameter. Moreover, the hashing consistency learning objective in the target domain is:

$$\mathcal{L}_{CL}^t = -\mathbb{E}_{\boldsymbol{x} \in \mathcal{B}^t} \log \frac{\exp\left(\hat{\boldsymbol{b}}_i \cdot \hat{\boldsymbol{b}}_k / \tau\right)}{\sum_{i=1}^{2|\mathcal{B}^s|} \exp\left(\hat{\boldsymbol{b}}_i^s \cdot \hat{\boldsymbol{b}}_{k'}^s / \tau\right)}. \tag{10}$$

Finally, the consistency learning objective is derived by combining both source and target domains:

$$\mathcal{L}_{CL} = \mathcal{L}_{CL}^s + \mathcal{L}_{CL}^t. \tag{11}$$

The introduction of consistency learning offers two primary benefits for hash code generation: (1) By maximizing the consistency of hash codes between similar images, our IDEA ensures the generation of similarity-preserving hash codes, which is essential for effective search engines. (2) By minimizing the similarity of hash codes derived from dissimilar images, our IDEA encourages hash codes to be uniformly distributed in the Hamming space, which enhances the capacity of each hash bit.

### 4.5 Invariance under Intervention

However, in practice, causal parts and non-causal-part could be still closely related. For example, waterbirds are typically paired with water backgrounds in the training set. To further reduce the potential hidden effects of non-causal parts, we introduce invariant learning [28, 70, 6], which first introduce a generative model to do interventions and then encourages hash codes to be sufficiently invariant to different non-causal parts.

In detail, a generative model $G(\cdot, \cdot)$ is first learnt to reconstruct original images using both causal and non-causal features. To optimize the generative model, the reconstruction loss in a mini-batch is written in the formulation of:

$$\mathcal{L}_{RE} = \mathbb{E}_{\boldsymbol{x}_i \in \mathcal{B}} ||\boldsymbol{x} - G(g^c(F(\boldsymbol{x}_i)), g^n(F(\boldsymbol{x}_i)))||_2^2. \tag{12}$$

With the well-trained model, we can simulate the intervention by utilizing different non-causal features. Here, we first generate synthetic images using $\boldsymbol{f}_i^c$ and $\boldsymbol{f}_k^n$:

$$\boldsymbol{x}_i^{+m} = \boldsymbol{x}_i | do(\boldsymbol{f}_m) = G(\boldsymbol{f}_i^c, \boldsymbol{f}_m^n), \tag{13}$$

where $do(\boldsymbol{f}_m^n)$ denotes to impose non-causal features from a different sample $\boldsymbol{x}_m^n$ in $\boldsymbol{B}$. Then we feed the synthetic sample into the hashing network to generate hash codes, $\boldsymbol{b}_i^{+m} = \psi(g^c(F(\boldsymbol{x}_i^{+m})))$ and then minimize the variance under different intervention. The objective can be written as:

$$\mathcal{L}_V = \mathbb{E}_{\boldsymbol{x}_i \in \mathcal{B}}[Var_m(\boldsymbol{b}_i^{+m})] = \mathbb{E}_{\boldsymbol{x}_i, \boldsymbol{x}_m \in \mathcal{B}} ||\boldsymbol{b}_i^{+m} - \bar{\boldsymbol{b}}_i||^2, \tag{14}$$

where $\bar{\boldsymbol{b}}_i = \mathbb{E}_{\boldsymbol{x}_m \in \mathcal{B}}[\boldsymbol{b}_i^{+m}]$ is the mean of synthetic hash codes. With effective invariant learning, we can generate domain-invariant binary codes for effective cross-domain retrieval.

**Algorithm 1** Training Algorithm of IDEA

---

**Require:** Source data $\mathcal{D}^s$, target data $\mathcal{D}^s$, code length $L$;
**Ensure:** The hashing network $\psi(g^c(F(\cdot)))$;
 1: Warm up our backbone by minimizing $\mathcal{L}_D$ and $\mathcal{L}_{RE}$;
 2: **repeat**
 3:     Sample a mini-batch $B^s$ and $B^t$ from $\mathcal{D}^s$ and $D^t$, respectively;
 4:     Generate two augmented views for each sample;
 5:     Calculate positive set by Eqn. 8;
 6:     Generate synthetic samples under intervention by Eqn. 13;
 7:     Calculate the final loss objective in Eqn. 15;
 8:     Reverse the gradients $T^n$;
 9:     Update parameters by gradient descent;
10: **until** convergence

---

Table 1: MAP performances on two bench-marking datasets with 64-bit hash codes.

| Methods | Office-Home | | | | | | Office31 | | | | | | Avg |
|---|---|---|---|---|---|---|---|---|---|---|---|---|---|
| | P2R | C2R | R2A | R2P | R2C | A2R | A2D | A2W | W2D | D2A | W2A | D2W | |
| *Unsupervised Hashing Methods* | | | | | | | | | | | | | |
| SH [62] | 15.03 | 8.77 | 12.87 | 16.13 | 8.24 | 13.71 | 12.02 | 9.83 | 34.72 | 11.28 | 9.85 | 34.37 | 15.57 |
| ITQ [17] | 26.81 | 14.83 | 25.37 | 28.19 | 14.92 | 25.88 | 29.55 | 28.53 | 58.00 | 26.83 | 25.09 | 58.89 | 30.24 |
| DSH [34] | 8.49 | 5.47 | 9.67 | 8.26 | 5.28 | 9.69 | 16.66 | 15.09 | 39.24 | 16.33 | 13.58 | 41.07 | 15.74 |
| LSH [16] | 12.24 | 6.94 | 11.45 | 13.45 | 7.24 | 11.49 | 16.04 | 15.35 | 38.80 | 13.60 | 14.67 | 43.99 | 17.11 |
| SGH [24] | 24.51 | 13.62 | 22.53 | 25.73 | 13.51 | 22.93 | 24.98 | 22.47 | 53.94 | 22.17 | 20.52 | 56.36 | 26.94 |
| OCH [35] | 18.65 | 10.27 | 17.54 | 20.15 | 10.05 | 18.09 | 24.86 | 22.49 | 51.03 | 22.45 | 20.79 | 53.64 | 24.17 |
| *Transfer Hashing Methods* | | | | | | | | | | | | | |
| ITQ+ [79] | 17.61 | 9.55 | 14.25 | - | - | - | 17.99 | 15.00 | 42.29 | - | - | - | 19.45 |
| LapITQ+ [79] | 16.89 | 10.37 | 13.56 | - | - | - | 19.96 | 18.24 | 43.32 | - | - | - | 20.39 |
| GTH-g [74] | 20.00 | 10.99 | 18.28 | 21.95 | 11.68 | 19.05 | 23.08 | 21.20 | 49.38 | 19.52 | 17.41 | 50.14 | 23.56 |
| DAPH [21] | 27.20 | 15.29 | 27.35 | 28.19 | 15.29 | 26.37 | 32.80 | 28.66 | 60.71 | 28.66 | 27.59 | 64.11 | 31.85 |
| PWCF [22] | 34.03 | 24.22 | 28.95 | 34.44 | 18.42 | 34.57 | 39.78 | 34.86 | 67.94 | 35.12 | 35.01 | 72.91 | 38.35 |
| DHLing [22] | 48.47 | 30.81 | 38.68 | 45.24 | 25.15 | 43.30 | 41.96 | 45.10 | 75.23 | 42.89 | 41.74 | 79.91 | 46.54 |
| PEACE [58] | 53.04 | 38.72 | 42.68 | 54.39 | 28.36 | 45.97 | 46.69 | 48.89 | 78.82 | 46.91 | 46.95 | 83.18 | 51.22 |
| Ours | **59.18** | **45.71** | **49.64** | **61.84** | **32.77** | **51.19** | **48.70** | **54.43** | **84.97** | **53.53** | **53.71** | **88.69** | **57.03** |

## 4.6 Summary

Finally, we summarize our framework with the following overall training objective as:

$$\mathcal{L} = \mathcal{L}_D + \mathcal{L}_{CL} + \mathcal{L}_{RE} + \mathcal{L}_V. \tag{15}$$

Moreover, the derivation of $sign(\cdot)$ is zero for any non-zero value, and therefore it is difficult to compatible with the gradient propagation. To tackle this challenges, we adopt $tanh(\cdot)$ to replace $sign(\cdot)$ during optimization, which produces approximate hash codes $\boldsymbol{v} = tanh(\psi(g^c(F(\boldsymbol{x}))))$ for training. Our model is first warmed up by the disentanglement module optimized by the reconstruction loss. Then we optimize the whole network with mini-batch stochastic gradient descent and the detailed progress of our IDEA can be found in Algorithm 1.

## 5 Experiment

### 5.1 Experimental Settings

**Baselines.** We adopt a variety of state-of-the-art approaches for performance comparison, including six unsupervised hashing approaches (i.e., SH [62], ITQ [17], DSH [34], LSH [16], SGH [24], OCH [35]) and seven transfer hashing approaches (i.e., ITQ+ [79], LapITQ+ [79], GTH-g [74], DAPH [21], PWCF [22], DHLing [64], and PEACE [58]). Some results of ITQ+ and LapITQ+ are omitted because prior information is not accessible for sample pair construction.

**Datasets.** Experiments are conducted on different benchmark datasets: *(1) Office-Home dataset* [57]: This dataset contains examples from four domains (Ar, Cl, Pr, Re), each with 65 object categories. Two domains are selected as the source and target, resulting in six transferable image retrieval tasks

Table 2: MAP performances on the Digits dataset with 64-bit hash codes.

| | MNIST2USPS | | | | | | USPS2MNIST | | | | | | |
| Code Length | 16 | 32 | 48 | 64 | 96 | 128 | 16 | 32 | 48 | 64 | 96 | 128 | Avg |
|---|---|---|---|---|---|---|---|---|---|---|---|---|---|
| *Unsupervised Hashing Methods* | | | | | | | | | | | | | |
| SH [62] | 15.56 | 13.67 | 13.80 | 13.45 | 13.35 | 12.95 | 15.59 | 14.35 | 14.22 | 13.57 | 12.92 | 12.96 | 13.87 |
| ITQ [17] | 13.05 | 15.57 | 18.54 | 20.12 | 23.12 | 23.89 | 13.69 | 17.51 | 20.40 | 20.30 | 22.79 | 24.59 | 19.46 |
| DSH [34] | 20.60 | 22.21 | 23.68 | 24.28 | 25.73 | 26.50 | 19.54 | 21.22 | 22.89 | 23.79 | 25.91 | 26.46 | 23.57 |
| LSH [16] | 12.40 | 13.54 | 15.89 | 16.01 | 18.54 | 20.44 | 12.76 | 14.86 | 14.77 | 16.89 | 16.32 | 19.67 | 16.01 |
| SGH [24] | 14.24 | 16.69 | 18.72 | 19.70 | 21.00 | 21.95 | 13.26 | 17.71 | 18.22 | 19.01 | 21.69 | 22.09 | 18.69 |
| OCH [35] | 13.73 | 17.22 | 19.59 | 20.18 | 20.66 | 23.34 | 15.51 | 17.75 | 18.97 | 21.50 | 21.27 | 23.68 | 19.45 |
| *Transfer Hashing Methods* | | | | | | | | | | | | | |
| ITQ+ [79] | 22.84 | 21.20 | 20.68 | 19.15 | 17.99 | 18.52 | - | - | - | - | - | - | 20.06 |
| LapITQ+ [79] | 24.26 | 24.03 | 23.76 | 24.59 | 23.33 | 22.73 | - | - | - | - | - | - | 23.78 |
| GTH-g [74] | 20.45 | 17.64 | 16.60 | 17.25 | 17.26 | 17.06 | 15.17 | 14.07 | 15.02 | 15.01 | 14.80 | 17.34 | 16.47 |
| DAPH [21] | 25.13 | 27.10 | 26.10 | 28.51 | 30.53 | 30.70 | 26.60 | 26.43 | 27.27 | 27.99 | 30.19 | 31.40 | 28.16 |
| PWCF [22] | 47.47 | 51.99 | 51.44 | 51.75 | 50.89 | 59.35 | 47.14 | 50.86 | 52.06 | 52.18 | 57.14 | 58.96 | 52.60 |
| DHLing [22] | 49.24 | 54.90 | 56.30 | 58.28 | 58.80 | 59.14 | 50.14 | 51.35 | 53.67 | 58.65 | 58.42 | 59.17 | 55.67 |
| PEACE [58] | 52.87 | 59.72 | 60.69 | 62.84 | 65.13 | 68.16 | 53.97 | 54.82 | 58.69 | 60.91 | 62.65 | 65.70 | 60.51 |
| **Ours** | **58.89** | **64.48** | **65.72** | **67.48** | **70.24** | **74.34** | **60.99** | **61.47** | **65.45** | **67.97** | **69.72** | **72.31** | **66.59** |

Figure 3: We use 128-bit Precision-Recall curves to evaluate the performance of our method on the Office-31 and Office-Home datasets.

(P2R, C2R, R2A, R2P, R2C, A2R). *(2) Office-31 dataset* [42]: This widely used benchmark dataset contains over 4000 examples classified into 31 classes, which are from three domains (Am, We, Ds). We randomly select two domains as the source and target, resulting in six transferable image retrieval tasks (A2D, A2W, W2D, D2A, W2A, D2W). *(3) Digits dataset*: We focus on MNIST [26] and USPS [23], each containing ten handwritten digits. Each sample is re-scaled to $16 \times 16$. We select one dataset as the source and the other as the target, resulting in two transferable image retrieval tasks (MNIST2USPS and USPS2MNIST).

**Implementation Details & Evaluation metrics.** We perform experiments on an A100-40GB GPU. 10% of the target examples are used as test queries and the remaining target and source samples are viewed as the training set. As for the database, source domain data is adopted in cross-domain retrieval while target domain data is adopted in single-domain retrieval. The hashing network is optimized using mini-batch SGD with momentum. The batch size is set to 36 and the learning rate is fixed as 0.001. We evaluate the retrieval performance using four common metrics: mean average precision (MAP), precision-recall curve, top-N accuracy curve, and top-N recall curve. MAP is the primary metric for evaluating retrieval accuracy, while the precision-recall curve provides an overall performance indicator at different recall levels. The top-N accuracy curve shows the accuracy for different numbers of retrieval instances. As the key metrics, MAP and precision-recall curve can reflect the overall performance of our model in retrieval tasks. We also analyze the top-N accuracy and recall curves to evaluate the performance at different number of retrieved results. These comprehensive evaluations allow us to have an in-depth understanding of the model's retrieval ability.

## 5.2 Empirical Results

**Performance Comparison.** We present the results of our experiments on several benchmark datasets in Tables 1 and Table 2, where we report the mean average precision (MAP) scores achieved by our IDEA and other state-of-the-art methods. Our analysis of the results leads to the following conclusions. On the Office-Home and Office31 datasets, our IDEA achieves an outstanding average MAP score of 57.03, which significantly outperforms the second-best method, PEACE, with an average MAP score of 51.22. Furthermore, IDEA achieves the highest MAP scores across all query types, including P2R, C2R, R2A, R2P, R2C, A2R, A2D, A2W, W2D, D2A, W2A, and D2W. On the

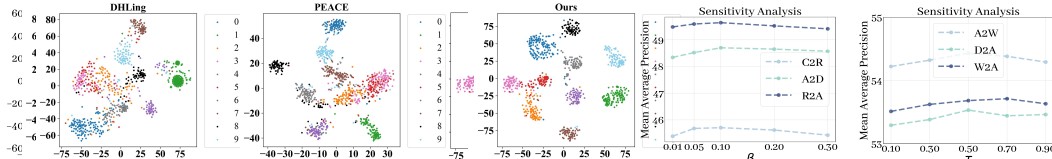

Figure 4: Visualization of 128-bit hash codes on MNIST dataset in left three columns; Sensitivity analysis results of our proposed IDEA on Office-Home and Office-31 datasets are shown in right two columns on three cross-domain retrieval tasks.

Digits dataset, IDEA consistently achieves the highest MAP scores for all hash code lengths, ranging from 16 to 128 bits. This demonstrates the consistency of our method across different code lengths. For instance, on the MNIST2USPS task, IDEA achieves an average MAP score of 74.34 for 128-bit codes, which is significantly higher than the second-best method, PEACE, with an average MAP score of 68.16. Overall, our experimental results demonstrate that the proposed IDEA is effective and outperforms state-of-the-art methods on various benchmark datasets. These findings highlight the potential of our method for practical applications in cross-domain image retrieval tasks. We present the recall-precision curves of four compared approches (PWCF, DHLing, PEACE, and IDEA) on two cross-domain retrieval tasks in Figure 3. Furthermore, in most cases, the curve of IDEA is higher than that of the other three baselines, indicating that the binary codes generated by IDEA works better for the hash table lookup strategy.

**Qualitative Results.** Here we provide a qualitative analysis of the model. Specifically, we use t-SNE visualization to display the hash codes learned by IDEA, PEACE, and DHLing (see the three subfigures on the left side of Figure 4). When compared to the PEACE and DHLing baseline methods, our IDEA-generated hash codes show more discriminative structures, with binary codes of different categories being better separated. These results indicate that the proposed method can produce hash codes with better discriminative power, which facilitates more effective image retrieval.

**Parameter Sensitivity.** In this part, we conduct sensitivity analysis for two important hyper-parameters, which is shown in the right subplot of the Figure 4. We can observe that the MAP values for all three cross-domain retrieval tasks slightly decrease as the $\beta$ parameter increases, indicating a negative impact on model performance. However, the decrease is not significant, allowing for an increase in $\beta$ to balance loss terms without causing a significant reduction in model performance. Similarly, the MAP values increase and then decrease as $\tau$ increases, demonstrating the potential for improved model performance. However, exceeding an optimal $\tau$ range may have a negative impact on retrieval performance. Thus, determining the optimal $\tau$ value through experimentation is necessary for achieving optimal retrieval performance in practical applications. In our experiments, we set $\beta$ and $\tau$ to 0.1 and 0.5, respectively.

**Ablation Study.** We have introduced several variants of our IDEA to investigate the impact of each components: (1) IDEA w/o $\mathcal{L}_{RE}$ removes the original image reconstruction loss from the generated model, (2) IDEA w/o $\mathcal{L}_{CL}$ removes the consistency learning loss, (3) IDEA w/o $\mathcal{L}_V$ removes the hash code variance under intervention, and (4) IDEA w/o $\mathcal{L}_{MI}$ removes the mutual information loss. By summarizing the

Table 3: Ablation Studies on six benchmark cross-domain retrieval tasks.

| Dataset | Office-Home | | Office31 | | Digits | |
|---|---|---|---|---|---|---|
| Method | C2R | R2A | A2D | A2W | U2M | M2U |
| IDEA w/o $\mathcal{L}_{RE}$ | 44.02 | 47.29 | 47.05 | 53.11 | 64.80 | 64.49 |
| IDEA w/o $\mathcal{L}_{CL}$ | 42.11 | 45.97 | 45.36 | 51.17 | 64.25 | 63.76 |
| IDEA w/o $\mathcal{L}_V$ | 45.10 | 49.09 | 48.21 | 53.96 | 57.34 | 66.97 |
| IDEA w/o $\mathcal{L}_{MI}$ | 44.89 | 48.92 | 48.05 | 53.79 | 67.12 | 66.85 |
| Ours | 45.71 | 49.64 | 48.70 | 54.43 | 67.97 | 67.48 |

results of these model variants in Table 3, We conclude, the ablation experiments show that removing any of the four components (i.e., $\mathcal{L}_{RE}$, $\mathcal{L}_{CL}$, $\mathcal{L}_V$, $\mathcal{L}_{MI}$) leads to a drop in performance on all datasets and tasks. Notably, $\mathcal{L}_{RE}$ and $\mathcal{L}_{CL}$ have the most significant impact. Our IDEA achieves the best or comparable performance on all datasets and tasks, demonstrating its effectiveness in learning disentangled representations for domain generalization. Specifically, it achieves the highest MAP on Office-Home and Digits datasets and the highest accuracy on the A2W task of the Office31 dataset, outperforming the ablated variants.

# 6   Conclusion

This work studies the problem of domain adaptive retrieval and proposes a novel method named IDEA to solve the problem. Our IDEA first disentangle causal and non-causal features within each image, guided by the principles of the information bottleneck. Moreover, these causal features are used to generate discriminative binary codes through consistency learning across both source and target domains. To generate domain-invariant binary codes, we simulates the intervention of various non-causal effects and encourage the invariance of hash codes. Extensive experiments on benchmark datasets validate the superiority of IDEA.

**Broader Impacts and Limitations.** This work improves the performance cross-domain retrieval, which can benefit real-world search engineers. Moreover, our work provide a new direction of incorporating invariant learning into cross-domain retrieval problems. One limitation of our work is that IDEA cannot tackle the open-set scenarios where target samples may come from unseen classes and we would extend our IDEA to more generalization scenarios in our future works. In particular, more advanced techniques such as modes in AICG and multimodal large-scale learning can be incorporated into our IDEA to improve the generalizability in a wider range of scenarios. In addition, we would explore more detailed visualization and interpretability analysis of invariant learning in the scope of cross-modal retrieval.

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

# A  The Upper Bound of Mutual Information

We define:

$$\hat{I}(\boldsymbol{F}^n, \boldsymbol{Y}) = \mathbb{E}_{p(\boldsymbol{F}^n, \boldsymbol{Y})}\left[\log p(\boldsymbol{y} \mid \boldsymbol{f}^n)\right] - \mathbb{E}_{p(\boldsymbol{F}^n)}\mathbb{E}_{p(\boldsymbol{Y})}\left[\log p(\boldsymbol{y} \mid \boldsymbol{f}^n)\right] \quad (16)$$

Then, we show that $\hat{I}(\boldsymbol{F}^n, \boldsymbol{Y})$ is an upper bound of $I(\boldsymbol{F}^n, \boldsymbol{Y})$. In formulation, we calculate their difference as follows:

$$\begin{aligned}
&\hat{I}(\boldsymbol{F}^n, \boldsymbol{Y}) - I(\boldsymbol{F}^n, \boldsymbol{Y}) \\
&= \mathbb{E}_{p(\boldsymbol{F}^n, \boldsymbol{Y})}\left[\log p(\boldsymbol{y} \mid \boldsymbol{f}^n)\right] - \mathbb{E}_{p(\boldsymbol{F}^n)}\mathbb{E}_{p(\boldsymbol{Y})}\left[\log p(\boldsymbol{y} \mid \boldsymbol{f}^n)\right] \\
&\quad - \mathbb{E}_{p(\boldsymbol{F}^n, \boldsymbol{Y})}[\log p(\boldsymbol{y} \mid \boldsymbol{f}^n) - \log p(\boldsymbol{y})] \\
&= \mathbb{E}_{p(\boldsymbol{F}^n, \boldsymbol{Y})}[\log p(\boldsymbol{y})] - \mathbb{E}_{p(\boldsymbol{F}^n)}\mathbb{E}_{p(\boldsymbol{y})}[\log p(\boldsymbol{y} \mid \boldsymbol{f}^n)] \\
&= \mathbb{E}_{p(\boldsymbol{Y})}\left[\log p(\boldsymbol{y}) - \mathbb{E}_{p(\boldsymbol{F}^n)}[\log p(\boldsymbol{y} \mid \boldsymbol{f}^n)]\right] \\
&= \mathbb{E}_{p(\boldsymbol{Y})}\left[\log\left(\mathbb{E}_{p(\boldsymbol{F}^n)}[p(\boldsymbol{y} \mid \boldsymbol{f}^n)]\right) - \mathbb{E}_{p(\boldsymbol{F}^n)}[\log p(\boldsymbol{y} \mid \boldsymbol{f}^n)]\right] \\
&\geq 0 \text{ (Jensen's Inequality)},
\end{aligned} \quad (17)$$

where the last inequality holds due to Jensen's Inequality with a convex function $\log(\cdot)$.

# B  The Details of Baselines

These baselines for model comparison are introduced as below:

- SH [62] is based on the theory of graph Laplacian. The binary code of a new data point is efficiently computed by the spectral analysis, enabling the out-of-distribution extension for unseen samples.
- ITQ [17] is an uncomplicated and effective alternative minimization algorithm. It can be incorporated into both supervised and unsupervised learning processes.
- LSH [16] is an important method in compressing high-dimensional data. It aims to map similar data points into the same bucket.
- DSH [34] is a model variant of LSH, which generates multiple view of the data point for metric learning using random projections.
- SGH [24] aims to compress high-dimensional data in a bit-wise manner, which is efficient for large-scale semantic similarity learning.
- OCH [35] leverage the tensor product to generate an ordinal graph, which guides the learning to generate more related hash codes.
- ITQ+ [79] is the early effort to implement transfer learning for hashing study, which learns a slack mapping from the source domain and achieves domain alignment through the quantization loss.
- LapITQ+ [79] is an extension of ITQ+ [79] by transferring the geometric relationship in the source domain to generate high-quality hash codes.
- GTH-g [74] selects the best hash mapping for target data using source data following the principle of maximum likelihood estimation.
- DAPH [21] utilizes a domain-invariant feature projection to reduce the domain discrepancy.
- PWCF [22] leverages a Bayesian model to learn discriminative hash codes and then infer the similarity structure using histogram features.
- DHLing [64] utilizes learning to cluster to optimize hash code within a single domain and then reduces the domain shift using the memory bank.
- PEACE [58] learns the semantics on target data using pseudo-labeling techniques and then minimizes the domain shift using both implicit and explicit manners.

# C  The Details of Datasets

Three popular benchmarks, i.e., Office-Home dataset, Office-31 and Digits, are adopted for performance comparison. Their details are elaborated as follows:

Table 4: MAP scores on Amazon → Dslr task with code lengths varying from 16 to 128 for single-domain retrieval.

| Bit | 16 | 32 | 48 | 64 | 96 | 128 |
|---|---|---|---|---|---|---|
| SH [62] | 30.54 | 35.66 | 40.84 | 42.50 | 44.01 | 45.64 |
| ITQ [17] | 40.83 | 49.27 | 53.92 | 56.16 | 59.31 | 59.41 |
| DSH [34] | 22.45 | 33.38 | 37.01 | 40.09 | 43.64 | 46.31 |
| LSH [16] | 16.04 | 26.18 | 34.36 | 39.68 | 45.60 | 49.04 |
| SGH [24] | 38.67 | 45.59 | 50.77 | 53.57 | 57.70 | 57.37 |
| OCH [35] | 33.30 | 41.65 | 48.18 | 50.78 | 50.06 | 53.74 |
| ITQ+ [79] | 35.03 | 42.62 | 41.85 | 43.12 | 39.82 | 39.12 |
| LapITQ+ [79] | 37.60 | 42.91 | 43.24 | 44.55 | 42.01 | 38.87 |
| GTH-g [74] | 37.11 | 45.69 | 50.67 | 50.22 | 54.37 | 55.81 |
| DAPH [21] | 46.74 | 49.43 | 56.10 | 58.63 | 59.82 | 60.41 |
| PWCF [22] | 49.94 | 53.05 | 58.26 | 59.08 | 60.11 | 62.35 |
| DHLing [64] | 52.08 | 56.43 | 59.92 | 60.17 | 61.19 | 63.44 |
| PEACE [58] | 55.43 | 57.89 | 60.23 | 61.21 | 62.30 | 64.14 |
| Ours | **61.25** | **62.65** | **65.14** | **67.06** | **68.20** | **70.04** |

Table 5: MAP scores on Product → Real task with code lengths varying from 16 to 128 for single-domain retrieval.

| Bit | 16 | 32 | 48 | 64 | 96 | 128 |
|---|---|---|---|---|---|---|
| SH [62] | 13.15 | 18.71 | 22.51 | 22.57 | 21.27 | 20.66 |
| ITQ [17] | 20.07 | 29.64 | 31.62 | 33.15 | 33.95 | 34.81 |
| DSH [34] | 6.10 | 11.44 | 12.21 | 16.61 | 16.46 | 14.45 |
| LSH [16] | 5.84 | 10.62 | 15.46 | 17.57 | 21.62 | 24.92 |
| SGH [24] | 18.97 | 26.18 | 30.61 | 32.61 | 33.82 | 34.97 |
| OCH [35] | 13.45 | 21.14 | 23.65 | 25.34 | 25.99 | 28.02 |
| ITQ+ [79] | 15.60 | 20.60 | 24.95 | 24.96 | 24.92 | 24.05 |
| LapITQ+ [79] | 16.78 | 22.26 | 22.76 | 22.29 | 22.33 | 21.85 |
| GTH-g [74] | 15.05 | 21.20 | 25.64 | 27.67 | 29.05 | 28.40 |
| DAPH [21] | 20.77 | 29.01 | 31.60 | 33.35 | 34.22 | 34.92 |
| PWCF [22] | 24.80 | 34.03 | 36.86 | 37.98 | 38.43 | 39.14 |
| DHLing [64] | 27.81 | 36.05 | 40.82 | 40.91 | 43.45 | 44.07 |
| PEACE [58] | 28.99 | 37.93 | 41.42 | 42.97 | 46.51 | 47.29 |
| Ours | **34.88** | **44.83** | **48.38** | **49.91** | **53.55** | **54.40** |

- Office-Home dataset [57] consists of image samples from four different domains, i.e., Art (A), Clipart (C), Product (P) and Real-World (R). There are 65 categories in the dataset. We select two domains as the source and target, respectively, which can produce six transferable image retrieval tasks (P2R, C2R, R2A, R2P, R2C, A2R).

- Office-31 dataset [42] is comprised of 4110 image samples from three different domains, i.e., Amazon (A), Webcam (W) and Dslr (D). Each domain corresponds to a platform where the images are collected. Similarly, we randomly select two domains as the source and target, which can result in six transferable image retrieval tasks (A2D, A2W, W2D, D2A, W2A, D2W).

- Digits consists of two datasets, MNIST [26] and USPS [23] and each contains handwritten digits from ten classes. The digits in both datasets are captured under different conditions. Here we consider two transferable image retrieval tasks (MNIST2USPS and USPS2MNIST).

# D   Additional Experimental Results

## D.1   Performance of Single-domain Retrieval

We first explore the results on single-domain retrieval, Table 4 compares the average mean average precision (MAP) of different hashing methods under different bit lengths for the single-domain retrieval task, i.e., Amazon → Dslr. The table is divided into two parts: the first part shows the performance of unsupervised hashing methods, including SH, ITQ, DSH, LSH, SGH, and OCH, while the second part shows the performance of transfer learning hashing methods, including ITQ+, LapITQ+, GTH-g, DAPH, PWCF, DHLing, and PEACE. Additionally, the results of IDEA method are highlighted with bold font and pink background, indicating that it achieves the highest MAP values across all hash code lengths. The table shows that IDEA performs better than other hashing methods for simplifying image descriptions and improving image retrieval efficiency. As the hash

Table 6: MAP scores on MNIST → USPS task with code lengths varying from 16 to 128 for single-domain retrieval.

| Bit | 16 | 32 | 48 | 64 | 96 | 128 |
|---|---|---|---|---|---|---|
| SH [62] | 46.30 | 47.82 | 48.58 | 49.12 | 49.27 | 47.81 |
| ITQ [17] | 13.39 | 22.58 | 33.12 | 39.67 | 40.80 | 40.16 |
| DSH [34] | 41.42 | 45.30 | 48.72 | 47.85 | 48.50 | 50.76 |
| LSH [16] | 26.69 | 33.49 | 33.21 | 35.64 | 38.92 | 38.53 |
| SGH [24] | 15.60 | 30.78 | 35.61 | 35.55 | 39.31 | 41.78 |
| OCH [35] | 24.23 | 32.90 | 36.10 | 36.34 | 39.95 | 44.36 |
| ITQ+ [79] | 50.22 | 49.66 | 46.64 | 44.38 | 44.53 | 43.21 |
| LapITQ+ [79] | 54.19 | 55.24 | 54.57 | 55.77 | 54.58 | 54.08 |
| GTH-g [74] | 45.41 | 39.72 | 36.66 | 34.34 | 32.11 | 34.73 |
| DAPH [21] | 47.53 | 54.86 | 57.36 | 60.15 | 60.65 | 60.39 |
| PWCF [22] | 50.21 | 49.41 | 57.27 | 60.06 | 64.71 | 64.00 |
| DHLing [64] | 51.25 | 50.48 | 58.44 | 63.13 | 65.09 | 67.02 |
| PEACE [58] | 52.77 | 56.25 | 61.03 | 65.27 | 67.71 | 69.99 |
| **Ours** | **60.81** | **63.32** | **68.05** | **72.11** | **74.47** | **76.73** |

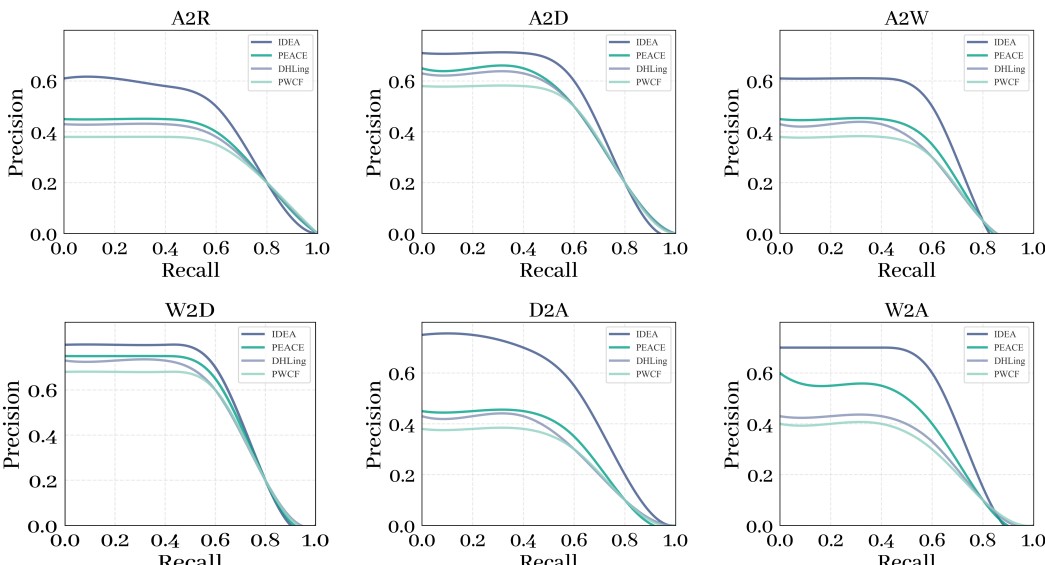

Figure 5: We use 128-bit Precision-Recall curves to evaluate the performance of our method on the Office-31 and Office-Home datasets.

code length increases, the MAP value of IDEA also increases, ultimately surpassing the performance of all other hashing methods at various hash code lengths. This indicates that IDEA has stronger feature extraction and hashing encoding capabilities and has significant practical value for image retrieval.

Table 5 compares the average MAP of different hashing methods under different bit lengths for the single-domain retrieval task, i.e., Product → Real. We can see that IDEA performs better than other hashing methods at all hash code lengths, especially at higher bit lengths such as 96, 128. Furthermore, the table highlights the results of IDEA in bold font and pink background, indicating that it achieves the highest MAP score at all bit lengths, namely 34.88, 44.83, 48.38, 49.91, 53.55, and 54.40. This further highlights the strong performances and advantages of IDEA, especially in learning compact hash codes and improving retrieval efficiency for single-domain retrieval tasks.

Table 6 compares the average MAP of different hashing methods under different bit lengths for the single-domain retrieval task MNIST → USPS. We can see that IDEA achieves the highest MAP values compared to other methods at all bit lengths, especially at higher bit lengths such as 96 and 128 bits, with MAP values of 74.47 and 76.73 respectively, far exceeding the MAP values of other methods. Additionally, the table highlights the results of IDEA in bold font and pink background, indicating that it achieves the highest MAP values at all bit lengths. Therefore, combining the information from the table, we can conclude that IDEA demonstrates outstanding performance in the

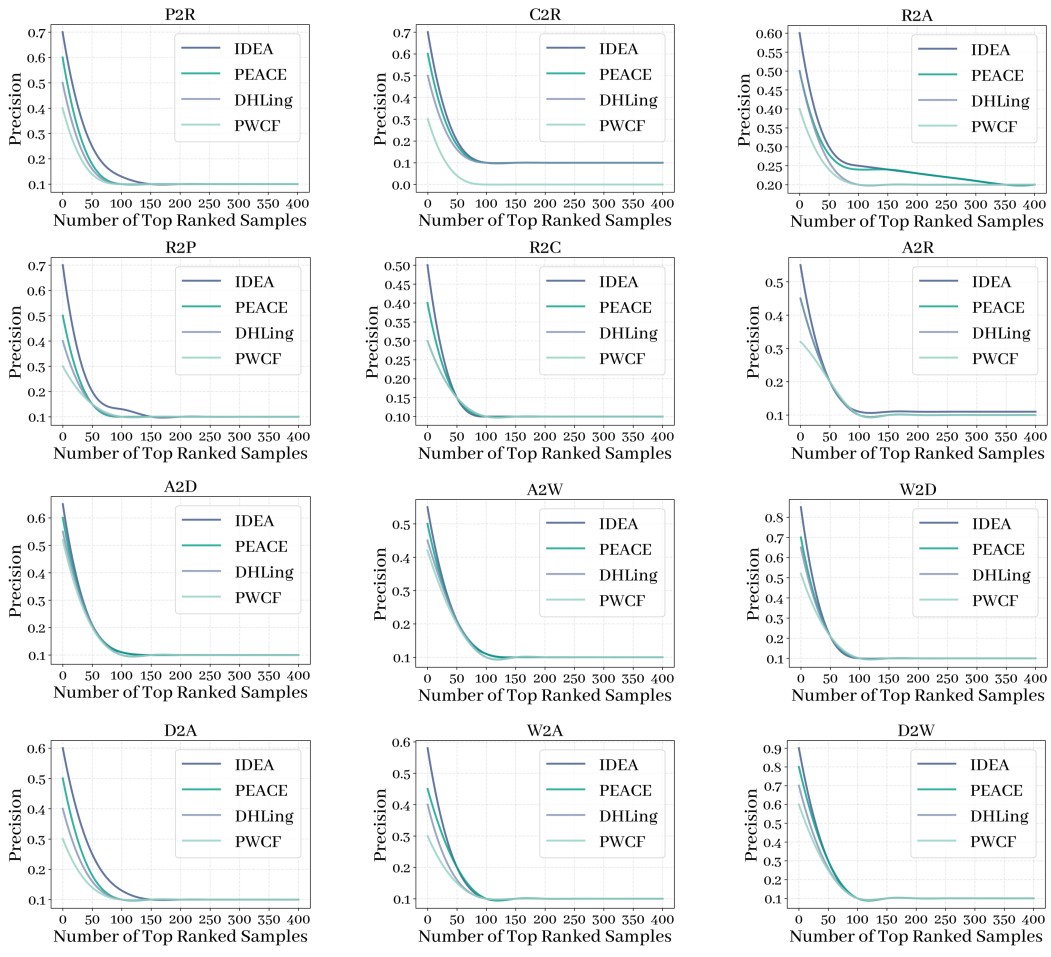

Figure 6: Top-N precision curves with 128 bits on Office-31 and Office-Home datasets.

single-domain retrieval task from MNIST → USPS, significantly enhancing retrieval efficiency and accuracy. This result suggests that IDEA can be a promising approach for improving image retrieval in similar domains.

## D.2 Performance of Precision-Recall Curves

Additional Precision-Recall curves for six image retrieval tasks, namely A2R, A2D, A2W, W2D, D2A, and W2A, can be found in this section. These tasks involve retrieving images from one domain and matching them to images from another domain. The Precision-Recall curves offer a visual representation of the balance between precision and recall for each task. As illustrated in Figure 5, these curves give insight into the effectiveness of our approach in dealing with different cross-domain retrieval scenarios. In particular, the curves highlight the strengths of our method in achieving high precision and recall values for each task. The supplementary material provides a detailed analysis of the experimental results, which further demonstrates the robustness and generalized ability of our approach. Overall, these additional results reinforce the effectiveness of our method in improving cross-domain image retrieval performance.

## D.3 Performance of Top-N Precision Curve and Top-N Recall Curve

The Top-N precision curve and Top-N recall curve depicted in Figure 6 and Figure 7, respectively, provide valuable insights into the performance of our IDEA algorithm compared to other methods in

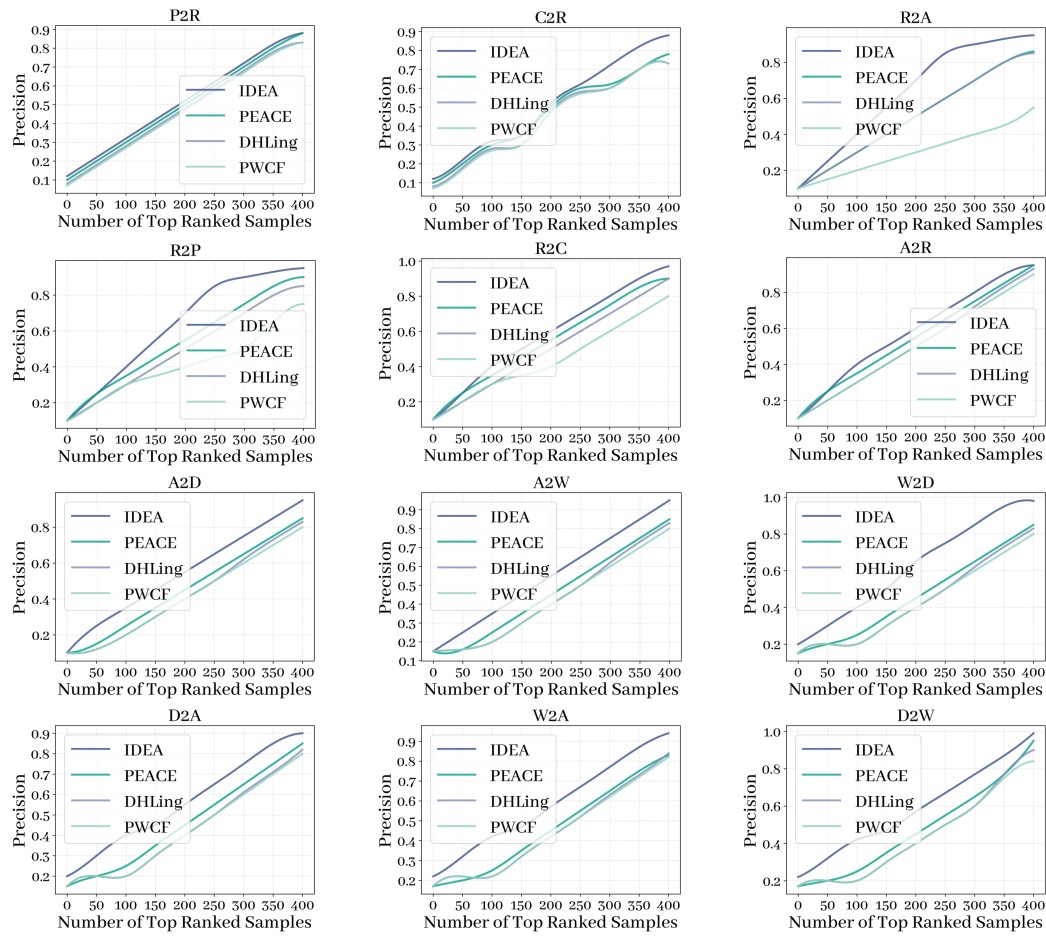

Figure 7: Top-N recall curves with 128 bits on Office-31 and Office-Home datasets.

image retrieval tasks. Based on the analysis of these curves, we can conclude that our IDEA algorithm outperforms other comparison methods. This is attributed to the fact that the calculation of the Top-N precision and Top-N recall curves is based on the Hamming distance ranking, and the IDEA algorithm achieves better performance under this metric. Therefore, from the perspective of evaluation metrics, the IDEA algorithm is superior to other comparison methods, as it offers superior precision and recall values for various cross-domain retrieval tasks. Our extensive experimental results further validate the effectiveness of our approach in improving the performance of cross-domain image retrieval.

## D.4 Ablation Study

We conduct more ablation studies on more datasets and the results are shown in Table 7. The results can validate our superiority again.

Table 7: Ablation Studies on three benchmark cross-domain retrieval tasks.

|  | Office-Home | | Office31 | |
| --- | --- | --- | --- | --- |
|  | Product2Real | Real2Product | Webcam2Dslr | Dslr2Webcam |
| IDEA w/o L_v | 58.23 | 60.92 | 83.88 | 87.22 |
| IDEA w/o L_MI | 57.87 | 60.46 | 83.71 | 87.32 |
| Ours | 59.18 | 61.84 | 84.97 | 88.69 |

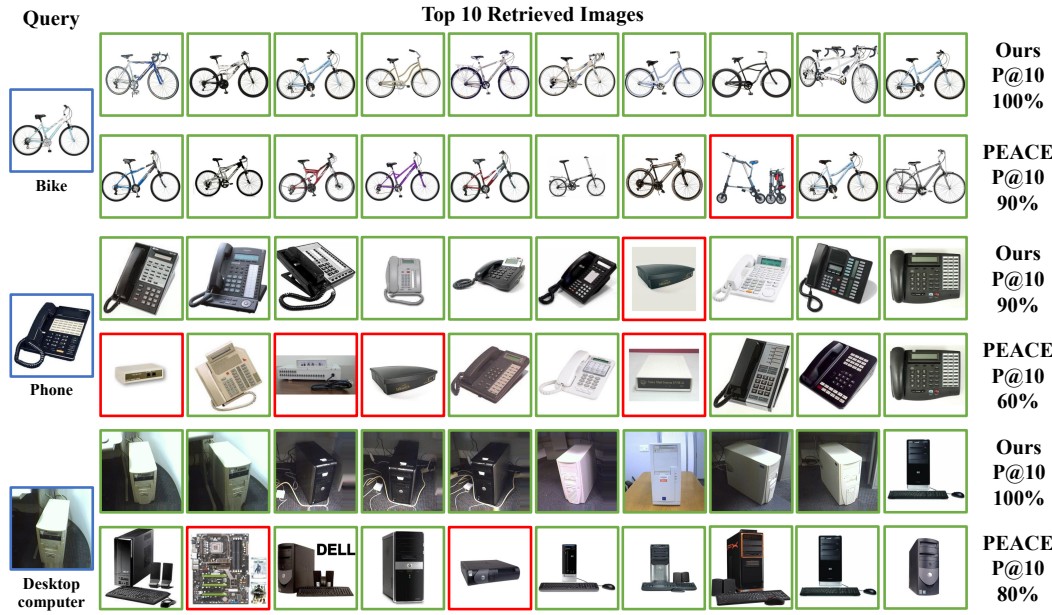

Figure 8: Top 10 Images and Precision@10 Examples on the Office-31 Dataset.

## D.5 Performance of Different Architectures

We compare the performance of both PEACE and IDEA with different network backbones, i.e., VGG-F and ViT. The compared results are shown in Table 8. From the results, we can have the following observations. Firstly, our IDEA can still achieve better performance under different architectures. Secondly, with strong networks, the performance of our IDEA can improve a lot.

Table 8: MAP scores with respect to different architectures.

| Dataset | Office-Home | | Office31 | | Digits | |
|---|---|---|---|---|---|---|
| | Clipart2Real | Real2Art | Amazon2Dslr | Amazon2Webcam | USPS2MNIST | MNIST2USPS |
| PEACE (VGG-F) | 38.72 | 42.68 | 46.69 | 48.89 | 60.91 | 62.84 |
| Ours (VGG-F) | 45.71 | 49.64 | 48.70 | 54.43 | 67.97 | 67.48 |
| PEACE (ViT) | 53.83 | 56.11 | 56.05 | 60.25 | 79.28 | 78.37 |
| Ours (ViT) | 58.56 | 64.39 | 63.27 | 66.30 | 83.39 | 82.96 |

## D.6 Case Study

Figure 8 demonstrates the top 10 retrieval results of the proposed IDEA and the best baseline method PEACE for three query samples on the Office-31 dataset. The blue box indicates the query, the green box indicates the correctly retrieved result, and the red box indicates the incorrectly retrieved result. The results demonstrate that IDEA performs better than PEACE in providing more relevant and user-desired examples, with the latter typically used for cross-domain retrieval. For instance, compared to PEACE, our IDEA returns more examples of the query "Phone," further validating its success in image retrieval tasks. Overall, these findings emphasize the effectiveness of our IDEA method in improving cross-domain image retrieval performance.

