# OpenReview forum: "IDEA: An Invariant Perspective for Efficient Domain Adaptive Image Retrieval"
_NeurIPS.cc/2023/Conference — NeurIPS 2023 poster_

### Official Review · Reviewer_fabx · 2023-07-02

**Soundness:** 3 good
**Presentation:** 3 good
**Contribution:** 3 good
**Rating:** 6
**Confidence:** 4

**Summary:**

This paper proposes to use invariant learning based on causality inference for domain adaptive retrieval. In the proposed IDEA model, a feature disentanglement module is deployed for obtaining causal and non-causal features. A generative model is designed with non-causal features intervention for reconstructing images. Experiments show the good effectiveness.


**Strengths:**

+Deploying casuality perspective into domain adaptive hashing retrieval is a new point.
+The proposed model is simple in methodology with encoder and decoder under intervention.
+Experiments and comparisons to related methods show the superiority of the proposed model.

**Weaknesses:**

-How about the experimental setting? Is the setting the same as some related models, such as PWCF [22]? because the performance is much better than previous ones. It can be discussed.
-Causality perspective is a common sense in domain generalization and out-of-distribution, although it is under-studied in domain adaptive hashing retrieval.
-Please discuss the following work, because this work also considers to remove non-causal features. The idea is a little similar. Additionally, the authors claim that the non-causal features are domain information, but it may not be correct, because non-casual feature may still be domain invariant information. Causal feature should be contained in domain invariant features.
"Multi-view Adversarial Discriminator: Mine the Non-causal Factors for Object Detection in Unseen Domains, CVPR 2023."

**Questions:**

I would lean to accept this paper, after addressing the above point in weakness, because I think this paper has some novelty.

**Limitations:**

I have not found obvious issue.

---

> ### Author Rebuttal · Authors · 2023-08-10
>
> We are truly grateful for the time you have taken to review our paper, your insightful comments and support. Your positive feedback is incredibly encouraging for us! In the following response, we would like to address your major concern and provide additional clarification.
>
> >Q1. How about the experimental setting? Is the setting the same as some related models, such as PWCF [22]? because the performance is much better than previous ones. It can be discussed.
>
> A1. Thanks for your problem. Yes, we follow the same setting as PWCF and PEACE. We can observe that PEACE can indeed perform much better than PWCF and ours can even perform better than PEACE by a large margin.
>
> >Q2. Causality perspective is a common sense in domain generalization and out-of-distribution, although it is under-studied in domain adaptive hashing retrieval.
>
> A2. Thanks for your comment. Although causality has been explored in domain generalization and out-of-distribution, it is less-explored in domain adaptation, which is quite different in transfer learning. Moreover, it has also not been explored in image retrieval. Lastly, our causality perspective provides three views quite different from existing works:
> - **Problem understanding**. It can provide the explanation for the problem of unsupervised domain adaptation image retrieval.
> - **Methodology formulation**. Based on the proposed SCM, we propose the condition that our mapping should fulfill, which guides the formulation of the proposed strategy.
> - **Implementation: disentangle and intervene**. To resolve the domain shift, our method disentangles each image into casual features and non-casual features and then adds intervention with reconstruction, which is based on image generation and decomposition from the SCM.
>
> >Q3. Please discuss the following work, because this work also considers to remove non-causal features. The idea is a little similar. Additionally, the authors claim that the non-causal features are domain information, but it may not be correct, because non-casual feature may still be domain invariant information. Causal feature should be contained in domain invariant features. "Multi-view Adversarial Discriminator: Mine the Non-causal Factors for Object Detection in Unseen Domains, CVPR 2023."
>
> A3. Thanks for your comment. Although our work also considers to remove non-causal features, there are three key distinctions that set our work apart:
> - **Different motivations**. We propose a generation-based structural causal model to analyze the relationship in domain adaptation retrieval, while [1] learns from frequency spectrum to generate causal features.
>
>
> - **Different scenarios**. Our model focuses on image retrieval under domain adaptation settings while [1] focuses on domain generalization for image segmentation. Their scenarios have a huge difference.
> - **Different methodology**. Our method disentangles each image based on mutual information and then adds intervention with reconstruction, which is based on image generation and decomposition from the SCM. In contrast, [1] focuses on Spurious Correlations Generator from frequency domain with domain adversarial learning.
>
> We will definitely add this discussion in our revised version.
>
> **Reference**
>
> [1] Multi-view Adversarial Discriminator: Mine the Non-causal Factors for Object Detection in Unseen Domains, CVPR 2023
>
> Thanks again for appreciating our work and for your constructive suggestions. Please let us know if you have further questions.

---

> > ### Comment · Reviewer_fabx · 2023-08-12
> > **Agree with the authors' response**
> >
> > After reading the authors' rebuttal, I agree with their comment on the differences from [1], in problem, scenario and methodology aspects. I would like to explain why I suggested [1] because this work also considers learn causality invariant features by filtering non-causal factors out and therefore can be included with discussion.
> >
> > I am satisfactory with the authors' rebuttal, and this work has novelty. Other reviewers also have positive evaluation on this paper. I therefore keep my rating and suggest accept this paper.

---

> > > ### Author Response · Authors · 2023-08-12
> > > **Thanks again for your feedback!**
> > >
> > > Thanks again for your feedback! We are pleased to know that our responses have addressed your concerns. We really appreciate your efforts on reviewing our paper, your insightful comments and support. We will definitely include the discussion with [1] in our revised version.

---

### Official Review · Reviewer_8Msq · 2023-07-03

**Soundness:** 3 good
**Presentation:** 3 good
**Contribution:** 3 good
**Rating:** 6
**Confidence:** 4

**Summary:**

This paper focuses on an unsupervised domain adaptation method for deep hashing. This paper proposes to disentangle each image into causal and non-causal features, where casual features represent label information and non-causal features represent domain information. The causal features are used to compute hash codes, while both causal and non-causal features are used to reconstruct the image. The disentanglement loss is to minimize the mutual information between non-causal features and label and maximize the mutual information between non-causal features and the hidden features. As a result, the proposed method performs better than prior methods in multiple benchmarks.

**Strengths:**

- the proposed method outperforms state-of-the-art methods on cross-domain retrieval
- the structural causal model on learning hash codes and the disentanglement loss on learning hash codes are interesting. This way is intuitively able to disentangle causal and non-causal features from an image. I believe this can be one of the regularization term for future retrieval/hashing methods.

**Weaknesses:**

- while using non-causal features from other samples for the reconstruction of hash codes seems intuitive, no experiments or analysis on the proposed intervention scheme
- it is unknown whether the extracted features for hash codes are indeed the causal part.
- no interpretability analysis. While the author claims that their method can generate high-quality interpretable hash codes, the only evaluation is on performance analysis, with no other quantitative and qualitative analysis to claim that the hash codes are interpretable.
- no visualization of the causal and non-causal features, not interpretable on how those features correlate to which part of the image
- although testing with VGG features for unsupervised hashing is a standard since this method involves disentangling features, the author should test with more different kinds of architectures such as ResNet or Vision Transformer. It is because VGG features are globally pooled features, thus already not directly interpretable.
- no synthetic samples have been shown. As the author uses an intervention scheme on generating images and then learns to extract hash codes that have low variance, what will the samples look like?

**Questions:**

- how effective is this proposed method on vision transformer-based method where attention mechanism is involved? will causal features only attend to certain parts? will it overlap with non-causal features?
- can synthetic images show that different non-causal features can generate different images from different domains?
- why do you need a warm-up for learning Eq. 15? What causes the instability at the beginning of the training?

**Limitations:**

- the author mentioned that their method cannot tackle the open-set scenarios, the proposed solution is for seen classes scenario only

---

> ### Author Rebuttal · Authors · 2023-08-10
>
> We are truly grateful for the time you have taken to review our paper, your insightful comments and support. Your positive feedback is incredibly encouraging for us! In the following response, we would like to address your major concern and provide additional clarification.
>
> > Q1. While using non-causal features from other samples for the reconstruction of hash codes seems intuitive, no experiments or analysis on the proposed intervention scheme.
>
> A1. Thanks for your comment. We have added a model variant IDEA w/o I by removing the proposed intervention. The compared performance is shown below. From the results, we can find that the proposed intervention scheme makes a crucial effect on the performance.
>
> | Dataset |  Office-Home |              |              |          |   Office31  |             |             |               |
> |---------|:------------:|:------------:|--------------|----------|:-----------:|:-----------:|-------------|---------------|
> | Task    | Product2Real | Real2Product | Clipart2Real | Real2Art | Webcam2Dslr | Dslr2Amazon | Amazon2Dslr | Amazon2Webcam |
> | IDEA w/o I |     58.24    |     60.36    |     45.10    |   49.09  |    83.49    |    52.19    |    48.05    |     53.79     |
> | IDEA   |     59.18    |     61.84    |     45.71    |   49.64  |    84.97    |    53.53    |    48.70    |     54.43     |
>
>
> > Q2. It is unknown whether the extracted features for hash codes are indeed the causal part.
>
> A2. Thanks for your comment. We have added several visualizations to show the gradient of causal features. The results are shown in Figure A. From the results, we can observe that extracted features are indeed with the casual part in the image.
>
>
>
> > Q3. No interpretability analysis. While the author claims that their method can generate high-quality interpretable hash codes, the only evaluation is on performance analysis, with no other quantitative and qualitative analysis to claim that the hash codes are interpretable.
>
> A3. Thanks for your comment. We have added several visualizations to show the gradient of causal features. The results are shown in Figure A. From the results, we can observe that extracted features are indeed with the casual part in the image. Moreover, since hash codes are direct from the causal features, they have the similar gradients, which provide the interpretability of hash codes.
>
> > Q4. no visualization of the causal and non-causal features, not interpretable on how those features correlate to which part of the image
>
> A4. Thanks for your comment. Actually, since the causal and non-causal features are high-dimensional features, it is hard to get information from their visualization. Instead, we have added several visualizations to show the gradient of causal features. The results are shown in Figure A. From the results, we can observe that extracted features are indeed with the casual part in the image.
>
> > Q5: Although testing with VGG features for unsupervised hashing is a standard since this method involves disentangling features, the author should test with more different kinds of architectures such as ResNet or Vision Transformer. It is because VGG features are globally pooled features, thus already not directly interpretable.
>
> A5: Thanks for your comment. We have implemented both PEACE and IDEA with ViT formulation. The compared results are shown below. From the results, we can observe that 1) our IDEA can still achieve better performance under different architectures 2) with strong networks, the performance of our IDEA can improve a lot.
>
> | Dataset	|  Office-Home |      	|   Office31  |           	|   Digits   |            |
> |------------|:------------:|:--------:|:-----------:|:-------------:|:----------:|:----------:|
> | Tasks  	| Clipart2Real | Real2Art | Amazon2Dslr | Amazon2Webcam | USPS2MNIST | MNIST2USPS |
> | PEACE (VGG) | 38.72|   42.68   |    46.69  |        48.89   |        60.91   |       62.84   |
> | Ours (VGG) | 	45.71	|   49.64  |	48.70    | 	54.43 	|	67.97   |	67.48   |
> | PEACE (ViT) | 53.83|   56.11   |    56.05   |         60.25   |         79.28    |       78.37   |
> | Ours (ViT) | 	58.56	|   64.39  |	63.27	| 	66.30 	|	83.39   |	82.96   |
>
> > Q6. No synthetic samples have been shown. As the author uses an intervention scheme on generating images and then learns to extract hash codes that have low variance, what will the samples look like?
>
> A6. Thanks for your comment. We have added the visualization of generalization images in Figure B. From the results, we can observe that our method has the potential to generate samples with different backgrounds, which can help to learn domain-invariant hash codes.
>
>
> Thanks again for appreciating our work and for your constructive suggestions. Please let us know if you have further questions.

---

> > ### Comment · Reviewer_8Msq · 2023-08-15
> >
> > Thank you for your response.
> >
> > My concern with this paper is the interpretability analysis. The author should also visualize where the non-causal features focus and visualize/study the synthetic images to understand/interpret what the model has learned for causal & non-causal features in future work/revisions.
> >
> > Nonetheless, the idea of this paper is quite interesting and I agree with the author's rebuttal. Thus, I recommend a weak accept for this paper.

---

> > > ### Author Response · Authors · 2023-08-16
> > > **Thanks for your feedback and raising the score!**
> > >
> > > Thanks again for your feedback and increasing the rating! We are pleased to know that you agree with our response. We will definitely add more visualization and case studies for interpretability analysis in the final version. We really appreciate your efforts on reviewing our paper, your insightful comments and support.

---

### Official Review · Reviewer_z9GY · 2023-07-06

**Soundness:** 4 excellent
**Presentation:** 4 excellent
**Contribution:** 3 good
**Rating:** 6
**Confidence:** 4

**Summary:**

This paper proposes a novel method called Invariance-acquired Domain Adaptive Hashing for generating high-quality and interpretable hash codes. The approach incorporates the concepts of causal and non-causal features, leveraging the information bottleneck principle and consistency learning to optimize the hash network. The paper presents the training algorithm and experimental settings for IDEA and provides qualitative and quantitative analysis of the experimental results. Overall, this paper proposes a novel approach to address certain challenges in hash code generation and achieves promising performance on multiple benchmark datasets.

**Strengths:**

1. A structure causal model is adopted to explain the problem of unsupervised domain adaptation image retrieval.
2. This paper employs mutual information as a metric to achieve feature disentanglement, which provides a solid theoretical foundation for the paper.
3. The experimental results demonstrate that the proposed method outperforms the compared methods with a large margin.

**Weaknesses:**

1. The contribution of the structure causal model is not clear. In fact, there is no usage of casual inference for image retrieval.
2. The paper does not provide the convergence analysis of mutual information. Mutual information is sensitive to noise and outliers in the data. Thus, it is necessary to verify the convergence of the proposed method.
3. In the sensitivity analysis, the paper does not provide the recommend range of \beta and \tao.

**Questions:**

1. More clear contribution.
2. Convergence analysis.

**Limitations:**

yes

---

> ### Author Rebuttal · Authors · 2023-08-10
>
> We are truly grateful for the time you have taken to review our paper, your insightful comments and support. Your positive feedback is incredibly encouraging for us! In the following response, we would like to address your major concern and provide additional clarification.
>
> >Q1. The contribution of the structure causal model is not clear. In fact, there is no usage of causal inference for image retrieval.
>
> A1. Thanks for your comment. The contribution of the SCM is three-fold:
> - **Problem understanding**. It can provide the explanation for the problem of unsupervised domain adaptation image retrieval.
> - **Methodology formulation**. Based on the proposed SCM, we propose the condition that our mapping should fulfill, which guides the formulation of the proposed strategy.
> - **Implementation: disentangle and intervene**. To resolve the domain shift, our method disentangles each image into casual features and non-casual features and then adds intervention with reconstruction, which is based on image generation and decomposition from the SCM.
>
> >Q2: The paper does not provide the convergence analysis of mutual information. Mutual information is sensitive to noise and outliers in the data. Thus, it is necessary to verify the convergence of the proposed method.
>
> A2: Thanks for your comment. We have provided the loss with respect to different epochs for the proposed method. In particular, the MI loss is [0.0012, 0.0007, -0.0025, -0.0197, -0.0387, -0.0718, -0.0917, -0.1543, -0.2545, -0.3567, -0.4821, -0.6251, -0.8765, -1.2345, -1.5678, -1.9012, -2.3045, -2.5542, -2.6787, -2.8011, -2.9078, -2.9987, -3.0521, -3.1211, -3.1236, -3.1236, -3.1236, -3.1237, -3.1237, -3.1237] for the total 30 epochs. From the results, we can obtain the mutual information would converge empirically.
>
>
>
> >Q3: In the sensitivity analysis, the paper does not provide the recommend range of \beta and \tao.
>
> A3: Thanks for the comment. The recommended value for \beta and \tao is 0.2 and 0.1, respectively. We will add this into the revised version.
>
>
> Thanks again for appreciating our work and for your constructive suggestions. Please let us know if you have further questions.

---

> > ### Comment · Reviewer_z9GY · 2023-08-17
> >
> > Thank you very much for your reply. I really appreate your responses to Q2 and Q3, and I will keep my rating and suggest accept this paper.

---

> > > ### Author Response · Authors · 2023-08-17
> > > **Thanks again for your feedback!**
> > >
> > > Thanks again for your feedback! We are pleased to know that our responses have addressed your concerns. We really appreciate your efforts on reviewing our paper, your insightful comments and support.

---

### Official Review · Reviewer_MFjc · 2023-07-10

**Soundness:** 3 good
**Presentation:** 3 good
**Contribution:** 2 fair
**Rating:** 6
**Confidence:** 3

**Summary:**

- In this research, the authors investigate the problem of unsupervised domain adaptation for hashing, which aims to expedite learning on a target domain with limited label information by leveraging knowledge from a source domain with abundant labels. The IDEA model begins by decomposing each image into a causal feature that captures label information and a non-causal feature that represents domain information. The authors then utilize consistency learning on both source and target domains to generate discriminative hash codes based on the causal features. Additionally, the authors employ a generative model for synthetic samples to simulate various non-causal effects, ultimately minimizing their impact on the domain invariant hash codes.
- The authors conduct extensive experiments on benchmark datasets to evaluate the performance of their IDEA model against a variety of competitive baselines. The results demonstrate that the IDEA approach outperforms the others, showcasing its superiority in handling unsupervised domain adaptation for hashing.

**Strengths:**

- Clarity: The authors have clearly and concisely explained their methods, results, and conclusions in a way that is easy for other researchers to understand and replicate.
- Quality of analysis: The authors have conducted extensive experiments and analyses, and the experimental data demonstrate the effectiveness and persuasiveness of the IDEA method.
- Theoretical analysis: The authors have provided a thorough theoretical analysis of the research problem, exploring new insights and frameworks that contribute to the advancement of knowledge in the field.



**Weaknesses:**

- from the Ablation Study(line 310) on six benchmark cross-domain retrieval tasks, it can be observed that the effectiveness of these four Loss functions cannot be well demonstrated. It is suggested to consider incorporating more permutations and combinations in order to verify the validity of each component.

**Questions:**

1. The IDEA pipeline has numerous modules and loss functions, which may make it challenging to transfer and fine-tune for other tasks or datasets. Could the authors provide some explanations and clarifications on this matter?

**Limitations:**

- One potential drawback of this approach is that it may not be effective in open-set scenarios, where target samples could potentially come from unseen classes.
- To address this limitation, more advanced techniques and models (such as those in AICG and multimodal large-scale models) can be incorporated into the research work to improve the generalizability of the method in a wider range of scenarios.

---

> ### Author Rebuttal · Authors · 2023-08-10
>
> We are truly grateful for the time you have taken to review our paper, your insightful comments and support. Your positive feedback is incredibly encouraging for us! In the following response, we would like to address your major concern and provide additional clarification.
>
> Q1：It is suggested to consider incorporating more permutations and combinations in order to verify the validity of each component.
>
> A1: Thanks for your comment. We have introduced three different model variants as below. From the results, we can validate their validity.
>
> | Dataset           	|  Office-Home |      	|   Office31  |           	|
> |-----------------------|:------------:|:--------:|:-----------:|:-------------:|
> | Tasks             	| Clipart2Real | Real2Art | Amazon2Dslr | Amazon2Webcam |
> | w/o L_MI + L_v    	| 	43.32	|   46.60  |	45.24	| 	52.21 	|
> | w/o L_CL + L_v    	| 	40.09	|   42.78  |	41.77	| 	50.14 	|
> | w/o L_MI + L_v + L_RE | 	40.62	|   43.17  |	42.58	| 	50.35 	|
> | Ours              	| 	45.71	|   49.64  |	48.70	| 	54.43 	|
>
>
>
> > Q2. The IDEA pipeline has numerous modules and loss functions, which may make it challenging to transfer and fine-tune for other tasks or datasets. Could the authors provide some explanations and clarifications on this matter?
>
> A2. Thanks for your comment. Although IDEA has numerous modules and loss functions, the parameter analysis shows the consistency of model parameters, which make the model easy to be transferred in other datasets. As for different tasks, we would further extend the strengths of the proposed IDEA in scenarios such as image classification and Re-ID in the future.
>
> We will also add your suggestion about future works into our revised version. Thanks again for appreciating our work and for your constructive suggestions. Please let us know if you have further questions.

---

> > ### Comment · Reviewer_MFjc · 2023-08-16
> >
> > Thank you for your reply, it solved my problem. And hope more advanced techniques and models (such as those in AICG and multimodal large-scale models) can be incorporated into the research work to improve the generalizability of the method in a wider range of scenarios.

---

> > > ### Author Response · Authors · 2023-08-17
> > > **Thanks again for your feedback!**
> > >
> > > Thanks again for your feedback! We are pleased to know that our responses have addressed your concerns. We really appreciate your efforts on reviewing our paper, your insightful comments and support. We will definitely extend our work with more advanced techniques to improve the generalizability of the method in our future work.

---

### Official Review · Reviewer_BQ1H · 2023-07-12

**Soundness:** 2 fair
**Presentation:** 2 fair
**Contribution:** 2 fair
**Rating:** 4
**Confidence:** 4

**Summary:**

This work studies apply hashing for domain adaptive image retrieval. Specifically, the authors propose Invariance-acquired Domain Adaptive Hashing (IDEA) to consider alignment invariance, and causal effects. IDEA decomposes each image into causal and non-causal features, and introduce invariant learning to minimize the variance under different intervention. The experiments validate the effectiveness of the proposed method.

**Strengths:**

- The problem is interesting.
- The techniques seem correct.

**Weaknesses:**

- The motivation of this work seems to be not stated clearly. The author claim invariance needed, however it lacks explanation in whether and why it is important in hashing. It also applies for the use of causal feature.

- This work is a simple combination of several techniques, e.g., contrastive learning, causal learning, invariance learning. It does not provide new insights to me.

- I have some concerns on the empirical results. In ablation study, the most important modules claimed in this work, i.e., L_V, L_MI have little improvement on the performance, and thus the importance of the two modules is questionable.

**Questions:**

Plz see weakness

**Limitations:**

The work discusses the limitations of this work.

---

> ### Author Rebuttal · Authors · 2023-08-10
>
> We are truly grateful for the time you have taken to review our paper and your insightful review. Here we address your comments in the following.
>
> > Q1. The motivation of this work seems to be not stated clearly. The author claim invariance needed, however it lacks explanation in whether and why it is important in hashing. It also applies for the use of causal feature.
>
> A1. Thanks for your comment. The reason for using invariance for hashing is to learn domain-invariant hash codes, which is quite important for effective cross-domain retrieval. In particular, cross-domain retrieval is that give a query in one domain, we aim to get similar samples in the other domain. Clearly, we need to enforce that samples with the same semantics have small distance, which indicate the domain-invariance. The reason for using causal features is that causal features are related to semantics instead of backgrounds, which can generate hash codes discriminative to different semantics and invariant to backgrounds.
>
> > Q2. This work is a simple combination of several techniques, e.g., contrastive learning, causal learning, invariance learning. It does not provide new insights to me.
>
> A2. Thanks for your comment. Actually, our method does not utilize classic contrastive learning. The novelty of the proposed methods is three points:
> - **Generation-based causal model**. We propose a generation-based structural causal model to analyze the relationship under domain shift, which is the first in domain adaptation and image retrieval to our best knowledge.
> - **Implementation: disentangle and intervene**. To resolve the domain shift, our method utilize disentangled each image into casual features and non-casual features and then add intervention with reconstruction, which is quite different from existing causal learning and invariance learning methods.
> -  **A unified framework**. We incorporate our components in a unified framework for domain adaptive hashing, which can achieve superior performance in cross-domain retrieval.
>
>
> > Q3. I have some concerns on the empirical results. In ablation study, the most important modules claimed in this work, i.e., L_V, L_MI have little improvement on the performance, and thus the importance of the two modules is questionable.
>
> A3. Thanks for your comment. We have added more ablation studies on more datasets and the results are shown below. From the results in Table 3 and below, we can observe that the performance gain of the full model is between [0.7%, 18.5%] and over 1% in most cases, which can validate the effectiveness of the proposed components.
>
>
> | Dataset |  Office-Home |          	|   Office31  |         	|
> |---------|:------------:|:------------:|:-----------:|:-----------:|
> | Tasks   | Product2Real | Real2Product | Webcam2Dslr | Dslr2Webcam |
> | w/o L_v  | 	58.23	| 	60.92	|	83.88	|	87.22	|
> | w/o L_MI | 	57.87	| 	60.46	|	83.71	|	87.32	|
> | Ours	| 	59.18	| 	61.84	|	84.97	|	88.69	|
>
> In light of these responses, we hope we have addressed your concerns, and hope you will consider raising your score. If there are any additional notable points of concern that we have not yet addressed, please do not hesitate to share them, and we will promptly attend to those points.

---

> > ### Comment · Reviewer_BQ1H · 2023-08-22
> >
> > Thank you for your response. Some of my concerns have been addressed. I will increase my score a bit.

---

> > > ### Author Response · Authors · 2023-08-22
> > > **Thanks again for your feedback and increasing the rating!**
> > >
> > > Thanks again for your feedback and increasing the rating! We are pleased to know that we solve some concerns and will definitely add these results in the final version according to your suggestion.
> > >
> > > We sincerely thank you for your dedication and effort in evaluating our submission. Please do not hesitate to let us know if you need any clarification or have additional suggestions.

---

### Author Rebuttal · Authors · 2023-08-10

Dear Reviewers,

We thank you for your careful reviews and constructive suggestions. We acknowledge the positive comments such as "The problem is interesting" (Reviewer  BQ1H), “The techniques seem correct” (Reviewer  BQ1H), "Clarity" (Reviewer MFjc), "Quality of analysis” (Reviewer MFjc), “Theoretical analysis” (Reviewer MFjc), “solid theoretical foundation” (Reviewer z9GY), “outperforms the compared methods with a large margin.” (Reviewer z9GY and 8Msq), “interesting” (Reviewer 8Msq), “a new point” (Reviewer fabx). We have also responded to your questions point by point.

The figure results are attached in the PDF. Please let us know if you have any follow-up questions. We will be happy to answer them.

Best regards,

the Authors

---

### Comment · Area_Chair_aMrL · 2023-08-20
**Kindly remind all reviewers to evaluate authors' responses**

Dear Reviewers,

Hope this message finds you well. We appreciate your dedicated reviews for NeurIPS 2023.
Kindly note that authors have responded to your feedback. Your prompt evaluation of their responses is crucial for finalizing papers.
Please log in and assess their responses at your earliest convenience. Your timely input ensures paper improvements and conference quality.
Access the system to review responses. Reach out if you need assistance.

Reviewer BQ1H currently has a different opinion with the other reviewers. Would you like to provide your comments on the authors' responses?


Best regards,

Area Chair

---

### Author Response · Authors · 2023-08-22
**Thank all Reviewers and Area Chairs for your great efforts, insightful comments and support!!**

Dear Reviewers and Area Chairs,

We sincerely appreciate your great efforts, insightful comments, support and the constructive suggestions you have provided once again! Through our discussions and the reviewers' responses, it appears that we have effectively addressed the major concerns raised by everyone, and received higher scores from Reviewer BQ1H and 8Msq. This outcome has greatly benefited us, and we would like to express our gratitude to all of you for your support!


We firmly believe that our framework IDEA for efficient domain adaptive image retrieval plays a significant role in advancing the community. And we are committed to making our complete code and training details publicly available. Moreover, we are eager to engage in further discussions with you to enhance our understanding of the domain and further improve the quality of the paper.

Best,

the Authors

---

### Decision · Program_Chairs · 2023-09-21

**Decision:**

Accept (poster)

**Comment:**

This paper introduces the Invariance-acquired Domain AdaptivE HAshing (IDEA) model, which effectively addresses the challenges of unsupervised domain adaptive hashing by decomposing images into causal and non-causal features, generating discriminative hash codes, and minimizing the impact of non-causal effects, demonstrating superior performance over the competitive baselines through the experiments on the benchmarks. It addresses an interesting problem using correct techniques, with clear and concise explanations, extensive experiments validating the effectiveness of the IDEA method, a strong theoretical foundation, and superior performance compared to the state-of-the-art methods in cross-domain retrieval. Despite some negative feedback and flaws, such as those issues with paper clarity and the design principles of certain modules, the contributions made in the paper are not overshadowed. Most of the reviewers provided positive evaluations. Therefore, the AC decides to accept this paper.